# Gas Permeation of Sulfur Thin-Films and Potential as a Barrier Material

**DOI:** 10.3390/membranes9060072

**Published:** 2019-06-14

**Authors:** Xicheng Jia, Thomas D. Bennett, Matthew G. Cowan

**Affiliations:** 1Department of Chemical and Process Engineering, University of Canterbury, Christchurch 8041, New Zealand; xicheng.jia@pg.canterbury.ac.nz; 2Department of Materials Science and Metallurgy, University of Cambridge, Cambridge CB3 0FS, UK

**Keywords:** sulfur, barrier materials, gas permeation, thin-films

## Abstract

Elemental sulfur was formed into poly(ether sulfone)-supported thin-films (ca. 10 µm) via a melt-casting process. Observed permeabilities of C_2_H_4_, CO_2_, H_2_, He, and N_2_ through the sulphur thin-films were <1 barrer. The sulfur thin-films were observed to age over a period of ca. 15 days, related to the reversion of polymerized sulfur to the S_8_ allotrope. This structural conversion was observed to correlate with an increase in the permeability of all gases.

## 1. Introduction

Elemental sulfur is produced as a by-product of petroleum refining and gas reserves using the hydrodesulfurization process [1]. Sulfur production is estimated at over 70 million tons each year [2], with over half the global supply produced in China, USA, Russia, Canada, and Saudi Arabia [3]. The majority of this sulfur is used in the production of sulfuric acid (more sulfuric acid is produced in the USA than any other chemical) [4,5], phosphate fertilizers [5,6], batteries [7], and in the vulcanization of natural and synthetic rubbers [8].

Elemental sulfur is usually present as a small cyclic molecule (S_8_), i.e., an 8-ring configuration, under ambient conditions. However, a variety of cyclic and polymeric allotropes also exist [9,10]. Depending on the allotrope mixture, “pure” sulfur can melt at temperatures between 112 and 130 °C. When liquid sulfur is heated above 159.4 °C, a discontinuity in physical properties is observed due to the polymerization of sulfur chains [9,10]. From the melt mixture, polymeric sulfur (yellow) and cyclic S_8_ rings (white) can be obtained depending on cooling conditions [9]. Above ca. 200 °C, sulfur takes on a dark red hue due to the presence of organic impurities or highly reactive S_3_, S_4_, and S_5_ species [9]. Upon cooling, sulfur slowly reverts to the thermodynamically stable S_8_ allotrope [11].

Sulfur is largely considered as a by-product, with commodity value, trading below US$68–109/ton [5]. This low value has resulted in efforts to derive value-added products from bulk-sulfur [10,12,13]. Since 2013, the “inverse vulcanization” process has shown promise in opening a synthetic route to sulfur-derived materials. Elemental sulfur in the liquid state is polymerized using the addition of vinyl monomers [14,15]. The resultant polymers are stable against recrystallization, can be self-healing [16,17,18], and have received considerable attention for application in battery electrodes [19,20,21]. Minor attention has been given to other applications such as the removal of mercury [22,23,24] and iron [25], anti-microbial materials [26], concrete and mortars [27], infrared optics [16,28,29,30], controlled release fertilizers [31], and oil spill clean-up [32].

The gas transport potential of sulfur-based materials remains, however, unexplored. As a dense material, thin-films of sulfur have potential to act as a barrier material. Key applications of barrier materials include low oxygen permeance to preserve the quality and taste of beverages and food, building construction, and hydrogen storage containers [33]. For barrier materials, the key materials–property goals are the ability to produce defect-free films, with low free volume [34], low molecule or polymeric chain mobility, and high density [33]. In general, materials with higher crystallinity make better barrier materials because the crystallite regions are impermeable—except along defects within the structure. However, high crystallinity is usually associated with brittleness and other non-desirable mechanical properties. There is one example of elemental sulfur being used in the synthesis of microporous polymers for application as adsorbents for natural gas sweetening [35]. However, our literature searching to date (Appendix A) suggests that the gas transport properties of elemental sulfur have not been characterized and it has not been considered as a barrier material. Herein, we present the gas permeability of sulfur thin-films when exposed to a selection of common gases (C_2_H_4_, CO_2_, H_2_, He, and N_2_) to provide baseline data for the comparison of sulfur-derived polymeric materials.

## 2. Materials and Methods

### 2.1. Materials and Instrumentation

All manipulations were carried out under an atmosphere inside a fume hood at a temperature of 21 °C. Sulfur (S_8_, sublimed flower, reagent grade, ≥99.5%) was purchased from B.D.H. Laboratory Chemicals (Poole, England). Supor^®^ hydrophilic poly(ether sulfone) disc filters (70% porous) with a diameter of 47 mm, were used as a support material and purchased from Pall Corporation (Port Washington, NY, USA). CO_2_, N_2_, H_2_, He, and C_2_H_4_ gas cylinders with purities of ≥99.99% were purchased from BOC (Auckland, New Zealand). The Powder X-ray diffraction (PXRD) patterns of the samples were recorded using a SmartLab X-Ray Diffractometer (Rigaku, Tokyo, Japan) equipped with a Co-Kα radiation source. The specific surface area was measured by N_2_ adsorption/desorption measurements at 77 K using a Gemini VI surface area and pore analyzer. Scanning electron microscopic (SEM) images were obtained using a JEOL 700F scanning electron microscope (JEOL Ltd., Tokyo, Japan), and the corresponding energy dispersive X-ray spectroscopy (EDX) elemental mapping of sulfur was obtained by the same device. Thermogravimetric analysis and differential thermal analysis (TGA/DTA) were performed using a STA 449 F3 model from NETZSCH (Selb, Germany) with the TGA/DTA programmed as follows: equilibrate to 50 °C, then ramp to 300 °C at 10 °C/min under an N_2_ environment.

### 2.2. Synthesis of the Sulfur Membrane

A mass of 0.5 g of sulfur powder was transferred to a glass dish with a diameter of 55 mm, which was pre-heated to 175 °C on a hot plate (IKA^®^ RCT basic, Staufen, Germany). After the sulfur powder melted and its color turned to dark red, the viscosity of the polymerized sulfur was observed to decrease. Thereafter, one side of the support material was briefly immersed within the sulfur melt and removed when a visually uniform layer of sulfur was generated over the support. The resultant sulfur-coated membrane was removed, and cooled for 10 min at room temperature, before being used for further experiments. SEM imaging showed that defect-free layers of sulfur of approximately 10 µm were routinely produced.

### 2.3. Single-Gas Permeability Test

Single-gas CO_2_, N_2_, H_2_, He, and C_2_H_4_ permeability measurements were performed using a dead-end filtration unit similar to those reported in the literature [36,37] and schematically shown in Figure 1. Experiments were performed at room temperature (21 °C) for 15 h, and each gas was tested in triplicate for each membrane sample. Between experiments, the apparatus and membrane were evacuated for 5 h at room temperature using a dynamic vacuum of <0.1 torr.

## 3. Results and Discussion

### 3.1. Check of Sulfur Physio-Chemical Properties

The aim of this study was to provide baseline data for the comparison of sulfur derived polymeric materials. Therefore, we aimed to measure the gas permeability of polymeric sulfur. Thermogravimetric analysis (TGA), differential thermal analysis (DTA), and the differential (DDTA) were measured to characterize the thermal properties of the sulfur used in this study (Figure 2) and ensure behavior conformed to that previously reported [9,10]. As expected, complex melt behavior was observed from ca. 103–150 °C, followed by a discontinuity at ca. 175 °C related to the polymerization of sulfur chains. Casting of sulfur membranes was therefore performed at temperatures ≥175 °C. It was noted that at these temperatures, the sulfur became considerably less viscous and relatively easy to cast [9,10].

### 3.2. Preparation of Thin-Film Sulfur Membranes

Free-standing thick films of sulfur were prepared by casting molten sulfur on glass, Teflon, and aluminum foil surfaces. The free-standing films could be masked using adhesive tinfoil. However, our attempts to measure the permeability of >250 µm thick sulfur membranes was frustrated by the low permeability of sulfur, resulting in membrane permeances too low to measure using our experimental set-up. In an effort to generate thinner films, we submerged porous supports of hydrophilic and hydrophobic poly(tetrafluoroethylene) and poly(ether sulfone) (PES) in molten sulfur. Of the studied supports, PES was found to be the most suitable support material, retaining both shape and physical integrity. This method generated films with thicknesses of 150 µm. These thicknesses were then reduced further by using a dip-coating method, where only the surface of the support was submerged in the molten sulfur. This proved successful for reproducibly, preparing membrane materials with cohesive defect-free active layers of ca. ≥10 µm within poly(ether sulfone) supports (Figure 3), as confirmed by scanning electron microscopy (SEM) cross-sections (Figure 4) and top and bottom views (Figure 5). Although fragile to handle, these films were sufficiently robust for masking with adhesive aluminum foil and permeability testing with ca. 1 atm transmembrane pressure. The films are brittle and prone to cracking if handled indelicately and more robust materials would be advantageous for applied research.

It has been well-documented that polymerized sulfur slowly reverts to the S_8_ species [9,10,38]. We used powder X-ray diffraction (PXRD) to monitor this progression over time. Baseline PXRD data for the blank supports, free-standing sulfur, and supported thin-films are shown in the ESI (Appendix A). The four peaks observed between 23–30° correspond to PXRD previously reported for polymeric sulfur [39]. As shown in Figure 6, the PXRD pattern of the sulfur films is largely consistent to ca. 10 days, after which changes in the PXRD pattern, such as the re-emergence of the peak at ca. 20° and development of fine structure between 30° and 60° show a change in the organization of the underlying material. As described in Section 3.3, these structural changes (likely associated with the reversion of polymerized sulfur to the S_8_ allotrope) were related to increases in permeability of the thin-films.

### 3.3. Single-Gas Permeation Results

Single-gas permeability measurements of sulfur thin-films were made using a dead-end filtration apparatus (Figure 1). The gas permeability of the first thin-film (thickness ca. 10 µm) sulfur membrane we studied was observed to creep over time (Table 1), with a major discontinuity occurring after 15 days. These results coincide with the creep observed in the PXRD results (Figure 6) and are likely linked to the reversion of polymeric sulfur to the S_8_ allotrope. The observed gas permeabilities were low, even after the structural discontinuity, and suggest the thin films remained (Table 1).

Both before and after the ca. 15-day discontinuity in gas permeability properties related to the reversion of polymeric sulfur to the S_8_ species, the permeability properties of sulfur thin-films were comparable to the common barrier materials cellophane (cellulose) and poly(tetrafluoroethylene) (PTFE) (Table 2). This suggests that if a mechanically robust form of polymeric sulfur could be generated, the resulting properties would be suitable for use as a commercial barrier material.

## 4. Conclusions

Thin films of sulfur have low permeabilities comparable to the common barrier materials cellophane and PTFE. The sulfur films were observed to suffer from aging linked to the reversion of polymeric sulfur to the thermodynamically stable S_8_ allotrope. This aging is associated with an increase in the permeability of all gases. This work provides base data to which further studies on S-derived low cost barrier materials may be compared. Due to the low commodity price of sulfur, there is promise that these materials could serve as low-cost barrier materials and we look forward to reporting those measurements in the near future.

## Figures and Tables

**Figure 1 membranes-09-00072-f001:**
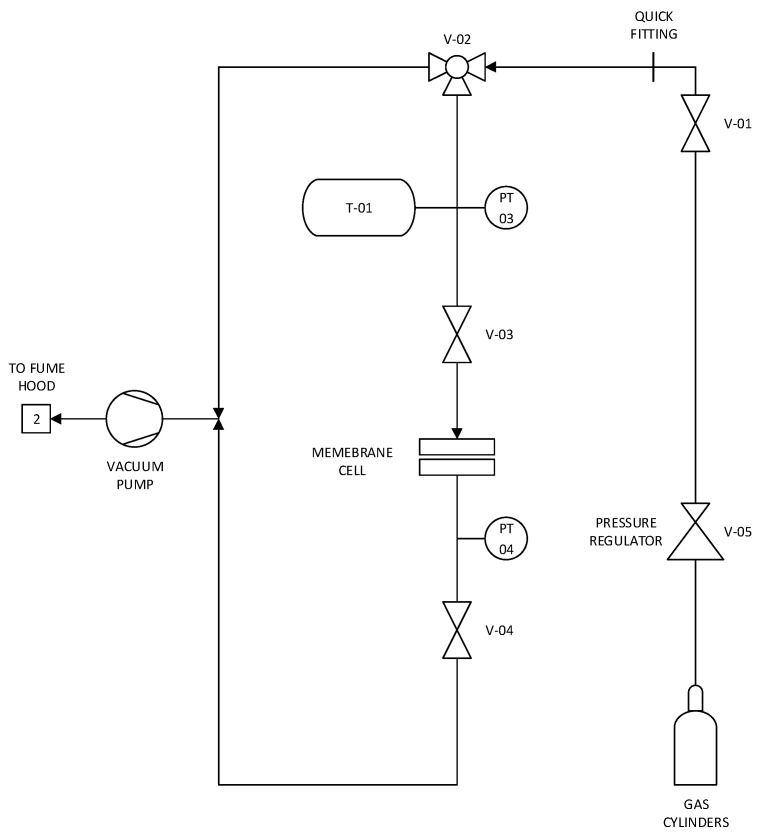
The dead-end filtration experimental set-up for measurement of single-gas permeability.

**Figure 2 membranes-09-00072-f002:**
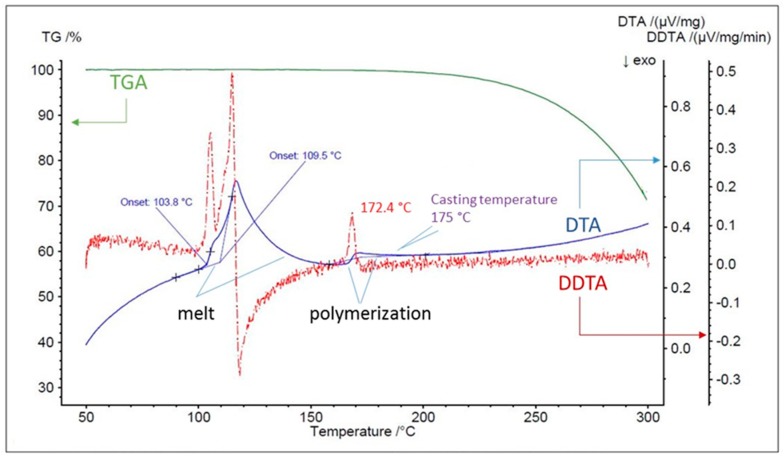
Thermogravimetric analysis (TGA), differential thermal analysis (DTA) and the differential (DDTA) spectra of the elemental sulfur up to 300 °C. (Green: TGA, Blue: DTA, and Red: DDTA).

**Figure 3 membranes-09-00072-f003:**
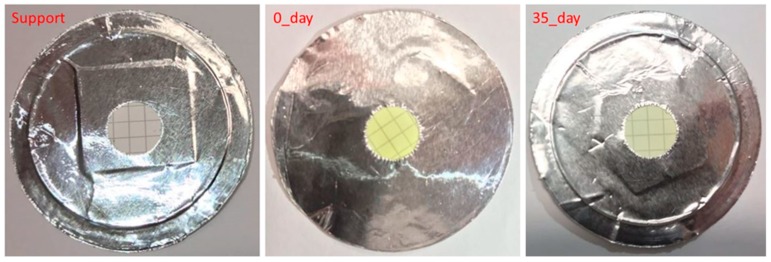
Images of the masked poly(ether sulfone) (PES) support (**left**), sulfur thin-film immediately after preparation (**middle**), and after membrane testing lasting ca. 35 days (**right**). Note the whitening of the aged membrane, suggestive of reversion of the polymeric sulfur to the S_8_ species.

**Figure 4 membranes-09-00072-f004:**
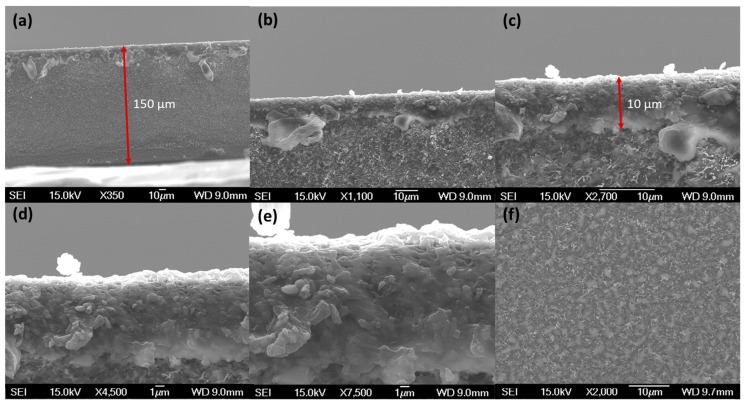
Scanning electron microscopic (SEM) images of cross sections of supported sulfur membranes prepared via dip coating: (**a**) full support with sulfur layer on top, (**b**) to (**e**) enlarged magnificatoin of ≥10 µm sulfur film, and (**f**) the empty PES support.

**Figure 5 membranes-09-00072-f005:**
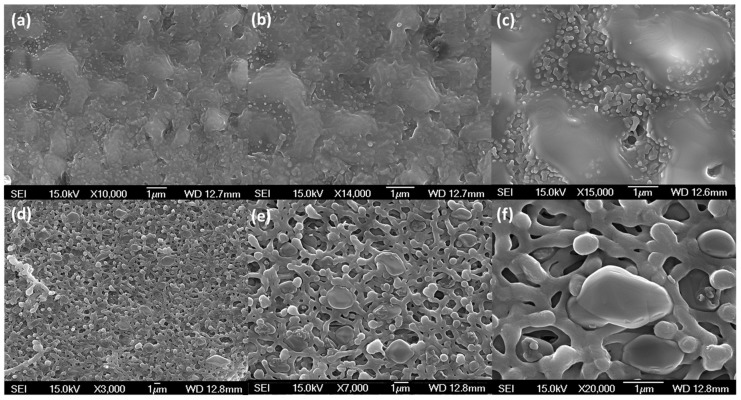
SEM images of the top and bottom surfaces of supported sulfur membranes prepared via dip-coating: (**a**–**c**) top-coating sulfur film and (**d**–**f**) open PES support.

**Figure 6 membranes-09-00072-f006:**
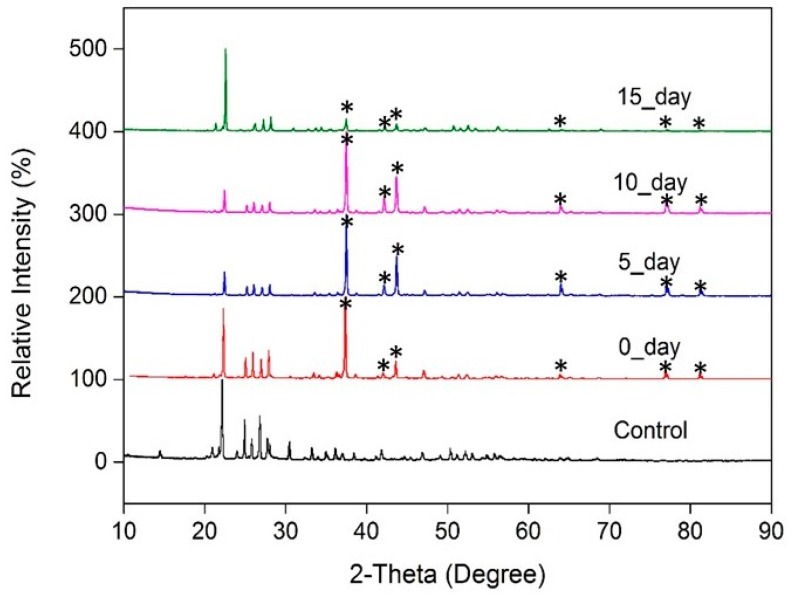
Powder X-ray diffraction (PXRD) patterns of the elemental sulfur (control), and sulfur supported-membranes for 0, 5, 10, and 15 days. Peaks with asterisk (*) are related to the porous support and the aluminium holder; control data are provided in the Appendix A.

**Table 1 membranes-09-00072-t001:** Single gas permeation for 3 runs.

Gas	Run 1		Run 2		Run 3	
	Barrer	10^−16^ mol·m/(m^3^·s·Pa)	Barrer	10^−16^ mol·m/(m^3^·s·Pa)	Barrer	10^−16^ mol·m/(m^3^·s·Pa)
C_2_H_4_	0.26	0.9	0.90^2^	3^2^	0.84^2^	2.8^2^
CO_2_	0.29	1.0	0.85^2^	2.8^2^	0.98^2^	3.3^2^
H_2_	1.7^2^	4.4^2^	1.6^2^	5.4^2^	1.3^2^	4.4^2^
He	0.55	1.8	0.50	1.7	1.56^2^	5.2^2^
N_2_	0.38	1.3	0.41	1.4	0.60^2^	2.0^2^

^1^ Room temperature; initial pressure is ca. 1 bar, trans-membrane pressure is ca. 1 bar. ^2^ Measurements recorded after the 15-day discontinuity.

**Table 2 membranes-09-00072-t002:** Gas permeability of sulfur compared to common barrier materials.

Gas	Sulfur		Cellophane	Poly(tetrafluoroethylene)
Reference	Pre-Discontinuity	Post-Discontinuity	[40]	[41]
H_2_	0.71	1.6	0.43	13.27 ± 4.84
He	0.55	1.7	1.7	
O_2_			0.14	
N_2_	0.38	0.6	0.21	1.64 ± 0.23
CO_2_	0.29	0.9	0.31	13.0 ± 2.4
C_2_H_4_	0.26	0.9		

^1^ The units for all values reported are 10^−16^ mol·m/(m^3^·s·Pa).

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
