# Peer review of "Gas Permeation of Sulfur Thin-Films and Potential as a Barrier Material"

_membranes, 2019, doi:10.3390/membranes9060072_

Round 1

Reviewer 1 Report

Dear authors

I am happy to recommend your manuscript for publication.

However, I have three minors comments on your manuscript:

 - the SEM images:  Fig 4a is it the support and the sulfur layer? Or only PES support? The legend is not very clear, it seems that it is only the support. But it looks like there is a layer coated on it...

 - on the XRD pattern, it would be great to show what each peak corresponds to. There is also mention of "peaks with asterisks" but no asterisks are visible on the Fig6.

- on table 1, only data with SI units (mol.m/(m3sPa) are useful as you are comparing them with literature.

Best regards

Author Response

Dear Reviewer 1,

We appreciate your rapid response to our paper and the constructive critical feedback you provided.

1. - the SEM images: Fig 4a is it the support and the sulfur layer? Or only PES support? The legend is not very clear, it seems that it is only the support. But it looks like there is a layer coated on it...

v  Thank you very much for pointing this out. The image of Fig. 4a is the PES support with the sulfur layer on top. We have revised the images with a higher resolution and the title in Fig. 4 accordingly. Section 3.2, Fig. 4:

2. - on the XRD pattern, it would be great to show what each peak corresponds to. There is also mention of "peaks with asterisks" but no asterisks are visible on the Fig6.

v  Thank you for the suggestion. We have revised Fig. 6 accordingly, Section 3.2. We have also included the phrase and reference on line 138: "The 4 peaks observed between 23-30° correspond to PXRD previously reported for polymeric sulfur [39].

3. - on table 1, only data with SI units (mol.m/(m3žsžPa) are useful as you are comparing them with literature.

v  Thank you for the suggestion. We included Barrer as it was commonly used in the past (and still widely used in some parts of the world) for gas permeation data. We include Barrers here to make the paper universally accessible.

Thank you again for your rapid and constructive comments.

Reviewer 2 Report

Jia, Bennett and Cowan report a study in the preparation of sulfur membranes and their gas permeability. There has been a resurgence in the synthesis and applications of sulfur-derived materials and this study adds a new and important dimension to this field. Specifically, the team prepared sulfur membranes on various supports and studied how the polymeric sulfur reverts back to S8 over time. Concurrently, the permeability to several gases was tested over the lifetime of these membranes. This study should be published and it is important for several reasons. First, it is an entirely new application of high-sulfur materials. Second, it is demonstrated that these sulfur membranes can be effective barriers to several gases, thereby opening opportunities in converting sulfur into protective and insulating films and related barriers.

I only have a few minor suggestions for the authors to consider:

How durable are the films? Are they brittle or damaged easily? It might be worth adding a bit more discussion about this issue.

The authors use the term ‘reverse vulcanization’ in the introduction, but it is probably better to keep with Pyun’s original descriptor ‘inverse vulcanization.’

The scale bars in the micrographs in Figures 4 and 5 are hard to see, please make these larger

Line 136 – patter to patern

In the caption for Fig 6 there is a reference to peaks with asterisks. I don’t see these in the XRD data? Thank you for providing the control XRD in the supporting information.

The authors do a nice job in the introduction highlighting some of the emerging uses of polymeric sulfur materials. The authors might consider adding these applications and references to the list:

Self-healing and repairable materials:

RSC Adv, 2018, 27892–27899

Adv Funct Mater, 2019, 1808989

Also re-cite reference 26 for this application

Controlled-release fertilisers:

2019 Org Biomol Chem, 2019, 1929–1936

Oil spill clean-up:

2018 Adv Sustainable Syst 1800024

Author Response

Dear Reviewer 2,

Thank you for your rapid and positive response to our work and the constructive feedback you have provided. We appreciate your interest in the work and note we are particularly interested in exploring the gas permeation of new materials, such as those in the self-healing references you suggested. Hopefully once this paper is published we will receive opportunities for collaboration from the wider community.

We address your specific comments below:

1. How durable are the films? Are they brittle or damaged easily? It might be worth adding a bit more discussion about this issue.

v This is definitely a point worth noting and will be a key advantage of sulphur materials produced through inverse vulcanization. We have included additional brief commentary on lines 120-123: "Although fragile to handle, these films were sufficiently robust for masking with adhesive aluminum foil and permeability testing with a ca. 1 atm transmembrane pressure. The films are brittle and prone to cracking if handled indelicately and more robust materials would be advantageous for applied research." 

2. The authors use the term ‘reverse vulcanization’ in the introduction, but it is probably better to keep with Pyun’s original descriptor ‘inverse vulcanization.’

Thank you very much for pointing this out. It has been corrected. Line 35.

3. The scale bars in the micrographs in Figures 4 and 5 are hard to see, please make these larger

v Absolutely. Higher quality images have been uploaded that include readable text.

4. Figure 6 has been replaced with asterisks included.

5. References have been included, thank you for bringing these papers to our attention.